# Phosphorylation-induced conformation of $\beta_2$-adrenoceptor related to arrestin recruitment revealed by NMR

Yutaro Shiraishi[1], Mei Natsume[1], Yutaka Kofuku[1], Shunsuke Imai[1], Kunio Nakata[2], Toshimi Mizukoshi[2], Takumi Ueda [1,3], Hideo Iwaï [4] & Ichio Shimada[1]

The C-terminal region of G-protein-coupled receptors (GPCRs), stimulated by agonist binding, is phosphorylated by GPCR kinases, and the phosphorylated GPCRs bind to arrestin, leading to the cellular responses. To understand the mechanism underlying the formation of the phosphorylated GPCR-arrestin complex, we performed NMR analyses of the phosphorylated $\beta_2$-adrenoceptor ($\beta_2$AR) and the phosphorylated $\beta_2$AR–$\beta$-arrestin 1 complex, in the lipid bilayers of nanodisc. Here we show that the phosphorylated C-terminal region adheres to either the intracellular side of the transmembrane region or lipids, and that the phosphorylation of the C-terminal region allosterically alters the conformation around M215[5.54] and M279[6.41], located on transemembrane helices 5 and 6, respectively. In addition, we found that the conformation induced by the phosphorylation is similar to that corresponding to the $\beta$-arrestin-bound state. The phosphorylation-induced structures revealed in this study propose a conserved structural motif of GPCRs that enables $\beta$-arrestin to recognize dozens of GPCRs.

[1] Graduate School of Pharmaceutical Sciences, The University of Tokyo, Hongo 7-3-1, Bunkyo-ku, Tokyo 113-0033, Japan. [2] Institute for Innovation, Ajinomoto Co., Inc, Kawasaki 210-8681, Japan. [3] Precursory Research for Embryonic Science and Technology (PRESTO), Japan Science and Technology Agency (JST), Chiyoda-ku, Tokyo 102-0075, Japan. [4] Research Program in Structural Biology and Biophysics, Institute of Biotechnology, University of Helsinki, P.O. Box 6500014 Helsinki, Finland. Correspondence and requests for materials should be addressed to I.S. (email: shimada@iw-nmr.f.u-tokyo.ac.jp)

G-protein-coupled receptors (GPCRs), which constitute the largest family of cell surface receptors in the human genome, are receptors for various hormones and neurotransmitters, and >30% of current drugs target GPCRs. Agonist binding to GPCRs leads to the activation of signal transduction, mediated by heterotrimeric G-proteins. The C-terminal region of activated GPCRs is phosphorylated by GPCR kinases (GRKs), and the phosphorylated GPCRs bind to arrestins. Arrestins bound to activated and phosphorylated GPCRs regulate the duration and strength of the cellular responses, by steric preclusion of the coupling between GPCRs and G-proteins, promotion of the GPCR internalization via clathrin-coated pits, and G-protein-independent, arrestin-mediated signal transductions. Activation of arrestins by GPCRs is required for cell homeostasis[1], and information about the arrestin activation mechanism will facilitate the development of drugs that selectively activate either the G-protein- or arrestin-mediated signal transductions, leading to reduced side effects. The phosphorylation of the C-terminal region of GPCRs by GRKs, as well as the agonist binding to GPCRs, is a critical step for the formation and activation of the GPCR–arrestin complex.

The tertiary structures of arrestins have been solved by X-ray crystallography[2–5]. In addition, the crystal structures of a complex between arrestin and a phosphorylated peptide that mimics the C-terminal region of a GPCR[6] and the arrestin 1 bound to rhodopsin[7, 8] have been solved. These structures indicated that the finger loop and N-domain of arrestin bind to the cytoplasmic cavity and the phosphorylated C-terminal region of GPCRs, respectively. However, structural information about the phosphorylated GPCRs is not available, although the structures of unphosphorylated GPCRs have been solved in various forms[9–11]. To understand the mechanism enabling the interactions of GPCRs with arrestin, the structures of the phosphorylated GPCRs must be clarified.

A lipid bilayer environment is reportedly preferable for GPCR phosphorylation by GRK[12]. It was recently reported that reconstituted high density lipoproteins (rHDLs), which are also known as nanodiscs[13], can accommodate membrane proteins within a 10-nm-diameter disc-shaped lipid bilayer[14]. Our NMR studies revealed that the conformational equilibria of $\beta_2$-adrenoceptor ($\beta_2$AR), one of the prototypical class A GPCRs, as well as those of other membrane proteins, more accurately reflect the physiological events when embedded in the lipid bilayer of rHDLs, as compared to in detergent micelles[15–17].

Here, we show the conformation of the transmembrane (TM) region and the C-terminal region of phosphorylated $\beta_2$AR embedded in the lipid bilayers of rHDLs by using NMR, together with the amino acid-selective and segment-selective isotope labeling methods[16, 18] (Fig. 1). This study identified a phosphorylation-induced conformation of $\beta_2$AR. On the basis of the conformation of the phosphorylated $\beta_2$AR, we will discuss the mechanism by which the activated and phosphorylated GPCRs are able to recognize $\beta$-arrestins.

## Results

**Preparation of the C-terminal segmentally-labeled $\beta_2$AR.** The C-terminal region of $\beta_2$AR possesses seven serine residues and four threonine residues, which are reportedly phosphorylated by GRKs (Fig. 1). For the observation of the NMR resonances from the $\beta_2$AR C-terminal region, the C-terminal region of $\beta_2$AR was segmentally labeled. The non-labeled TM region of $\beta_2$AR (1−348), expressed in an insect cell expression system, was fused with the labeled C-terminal region (349−413), expressed in an *E. coli* expression system, using intein-mediated protein *trans*-splicing (PTS)[19, 20] (Fig. 2a). Sodium dodecyl sulfate-polyacrylamide gel

electrophoresis (SDS–PAGE) analyses revealed that the PTS reaction proceeded with ~80% efficiency (Supplementary Fig. 1a). The segmentally labeled $\beta_2$ARs were reconstituted into the lipid bilayers of rHDLs. The size exclusion chromatography analysis revealed that the purified $\beta_2$ARs in rHDLs are monodisperse, with a Stokes diameter of 11 nm, in good agreement with the previously reported rHDL size[13] (Supplementary Fig. 1b).

For the evaluation of the $\beta$-arrestin signaling, the phosphorylation of $\beta_2$AR in rHDLs by GRK2, which is the main GPCR kinase of $\beta_2$AR, was analyzed, according to the previous report that the relative activity of the ligand to induce the phosphorylation of a GPCR correlates well with the efficacy of the $\beta$-arrestin signaling[21]. SDS-PAGE analyses of rHDL-embedded $\beta_2$AR phosphorylated by GRK2 using Pro-Q® Diamond staining, which stains phosphorylated proteins, revealed that, in the presence of the full agonist, the amount of phosphorylated $\beta_2$AR was remarkably larger than that in the presence of the inverse agonist, carazolol (Supplementary Fig. 1c, Supplementary Fig. 9). Therefore, the obtained $\beta_2$AR retains the activity to stimulate arrestin-mediated signaling in a ligand-dependent manner.

**Conformation of the phosphorylated C-terminal region.** In the $^1$H-$^{15}$N heteronuclear single-quantum coherence (HSQC) spectra of $\beta_2$AR, in which the 63 residues in the C-terminal region were segmentally labeled with $^2$H, $^{13}$C, and $^{15}$N, with or without the phosphorylation by GRK2, signals apparently originating from the 63 residues in the C-terminal region were observed (Fig. 2b and Supplementary Fig. 2b). Hereafter, $\beta_2$AR, in which

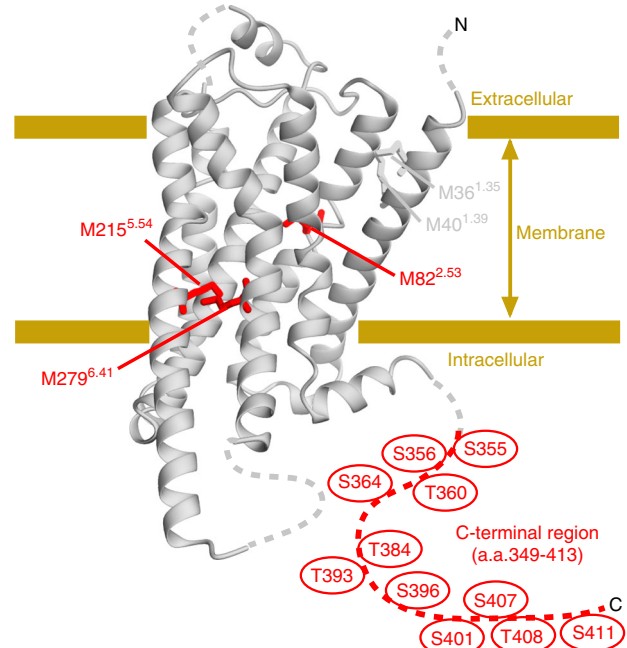

**Fig. 1** Distribution of the NMR probes on the structure of $\beta_2$AR. The crystal structure of $\beta_2$AR (PDB ID: 3SN6) is shown as a ribbon model. The N-terminal region, intracellular loop 3, and the C-terminal region, which are either substituted with T4 lysozyme, not observed, or truncated, are shown with dotted lines. The methionine residues observed for the analysis of the conformation of the TM region are shown as sticks. M82$^{2.53}$, M215$^{5.54}$, and M279$^{6.41}$, with resonances that reportedly exhibit remarkable chemical shift change upon activation, are shown as red sticks. The C-terminal region, which was segmentally labeled for the NMR observation, is shown by a red dotted line. The putative phosphorylated residues in the C-terminal region are shown. The structural model was prepared with Cuemol (http://www.cuemol.org/)

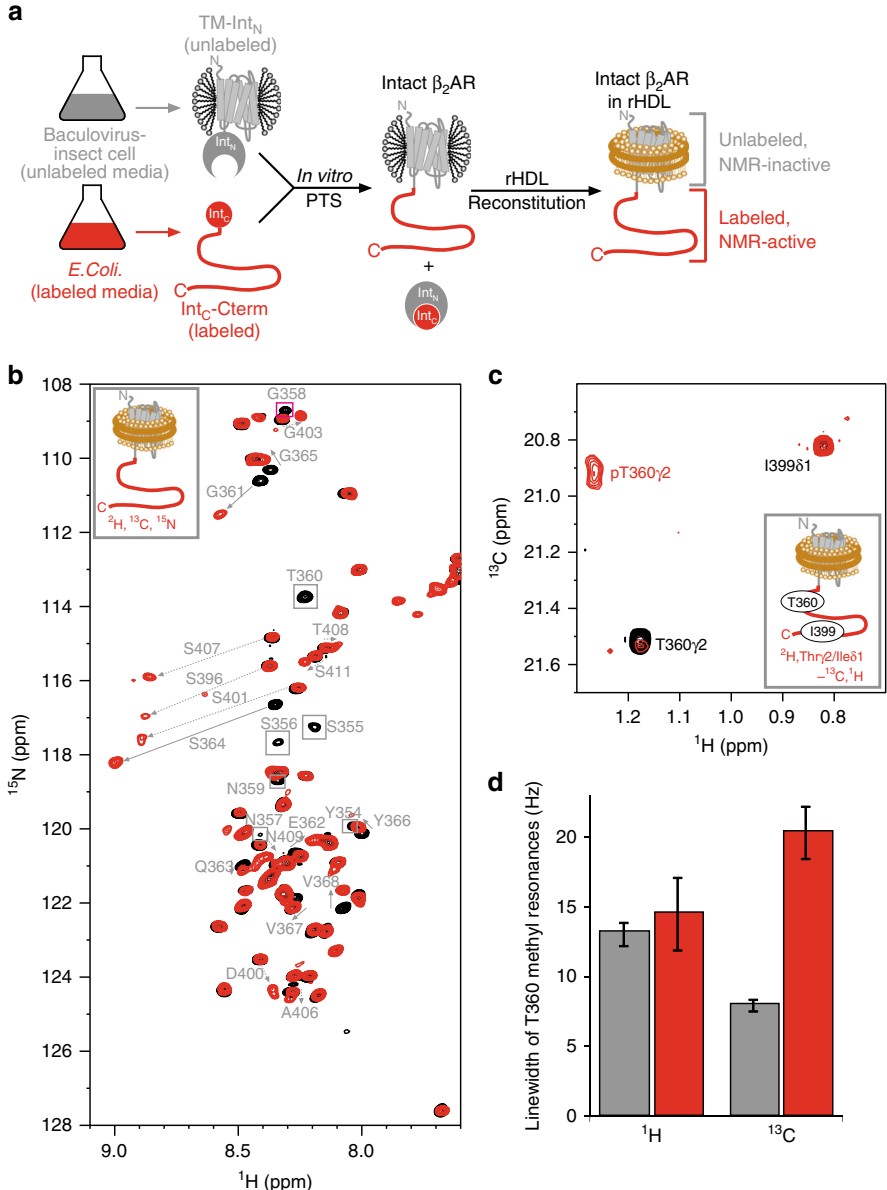

**Fig. 2** NMR spectra of the C-terminal region of the segmentally labeled $\beta_2$AR. **a** Schematic diagram of the preparation of segmentally labeled $\beta_2$AR. **b** Overlay of the $^1$H-$^{15}$N HSQC spectra of the unphosphorylated (black) and phosphorylated (red) {Cterm- [$^2$H, $^{13}$C, $^{15}$N]} $\beta_2$AR in rHDLs. Resonances that were not observed in the phosphorylated $\beta_2$AR are indicated with boxes, and resonances that exhibited different chemical shifts between the unphosphorylated and phosphorylated spectra are indicated with arrows. **c** Overlay of the $^1$H-$^{13}$C HMQC spectra of {Cterm- [$^2$H, Thr$\gamma$2, Ile$\delta$1-[$^{13}$C,$^1$H]]} $\beta_2$AR in rHDLs. **d** $^1$H and $^{13}$C linewidths of the T360 methyl signals of the unphosphorylated (gray bars) and phosphorylated (red bars) $\beta_2$AR in rHDL. The linewidths were calculated by fitting the Lorentzian function to the signals using Topspin 3.1. The error bars indicate the standard errors of the fit

C-terminal region was labeled with $^2$H, $^{13}$C, and $^{15}$N, is referred to as {Cterm-[$^2$H,$^{13}$C,$^{15}$N]} $\beta_2$AR. The resonance assignments of the spectra with or without the phosphorylation were accomplished by standard triple resonance experiments for {Cterm-[$^2$H,$^{13}$C,$^{15}$N]} $\beta_2$AR-rHDL and comparisons with the spectra of the peptide corresponding to the C-terminal region of $\beta_2$AR (Supplementary Fig. 2b). On the basis of comparisons between the spectra of the unphosphorylated $\beta_2$AR and those of the phosphorylated $\beta_2$AR, the degrees of the GRK2-mediated phosphorylation of each serine and threonine residue in the C-terminal region were calculated (Supplementary Table 1 and Supplementary Note 1).

In the unphosphorylated state, all resonances derived from the C-terminal region of $\beta_2$AR were sharp and exhibited small amide proton chemical shift dispersion, and the $^{13}$C$\alpha$ chemical shift

differences from those of a random coil were <1.6 ppm (Supplementary Fig. 3). Therefore, in the unphosphorylated state, the C-terminal region of $\beta_2$AR is highly flexible and does not form a specific secondary structure[22]. In the phosphorylated state, the amide resonances derived from G361−L413 were sharp and the $^{13}$C$\alpha$ chemical shift differences from those of a random coil were <1.6 ppm (Fig. 2b and Supplementary Fig. 3), suggesting that these residues are also flexible even in the phosphorylated state. In contrast, the amide resonances derived from G353−T360 could not be observed (Fig. 2b). Hereafter, the G353−T360 region is referred to as the "TM-proximal region" and the rest of the C-terminal region, G361−L413, is referred to as the "TM-distal region".

Resonances from methyl groups can be observed with higher sensitivity than those from amide groups, and the TM-proximal

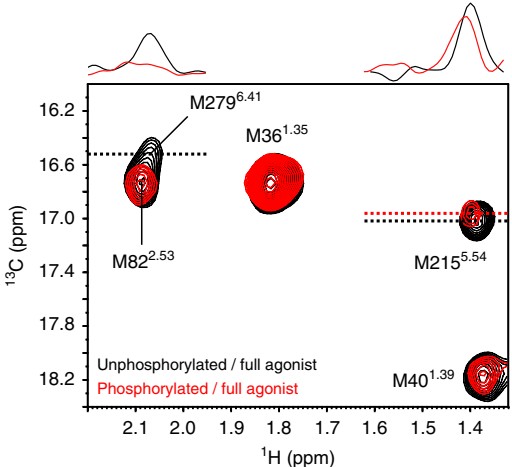

**Fig. 3** Conformation of the TM region of phosphorylated $\beta_2$AR. Overlay of the $^1$H-$^{13}$C HMQC spectra of [$^2$H-9AA, $\alpha\beta\gamma$-$^2$H, methyl-$^{13}$C-Met] $\beta_2$AR in rHDLs in the unphosphorylated state (black) and phosphorylated state (red). Cross-sections of the resonances from M215$^{5.54}$ and M279$^{6.41}$ are shown above the spectra

region contains a single methyl group at T360. Therefore, we utilized the T360γ2 methyl group to investigate the conformation of the TM-proximal region of the phosphorylated $\beta_2$AR. To overcome the signal overlaps, we mutated the threonine residues in the TM-distal region, and utilized a methyl group of I399 to probe the conformation of the TM-distal region. In the $^1$H-$^{13}$C HMQC spectra of the {Cterm- [$^2$H, Thrγ2/Ileδ1-$^{13}$C$^1$H$_3$]} $\beta_2$AR mutant, resonances corresponding to T360γ2 and I399δ1 were observed (Fig. 2c). In the phosphorylated state, the chemical shift of the I399δ1 methyl resonance was almost identical to that in the unphosphorylated state. Interestingly, the T360γ2 methyl resonance in the phosphorylated state exhibited remarkable $^{13}$C line broadening (Fig. 2d). The $^{13}$C chemical shifts of the threonine methyl signals are reportedly dependent on the sidechain $\chi^1$ angles[23]. Therefore, the TM-proximal region of the C-terminal region of phosphorylated $\beta_2$AR is not fully disordered, and undergoes exchange among multiple conformations that adopt different $\chi^1$ angles of T360, on the order of microseconds to milliseconds.

**Conformation of the TM helices of phosphorylated $\beta_2$AR.** To investigate the conformation of the TM region of phosphorylated $\beta_2$AR, we observed the NMR resonances from the methionine methyl groups in this region of $\beta_2$AR in the phosphorylated state. We prepared the phosphorylated [$^2$H-9AA, $\alpha\beta\gamma$-$^2$H-, methyl-$^{13}$C-Met] $\beta_2$AR 4Met mutant, in which four solvent- or lipid-exposed methionine residues were mutated, as previously described[16], embedded in rHDLs. $\beta_2$AR/4Met possesses five methionine residues: M36$^{1.35}$, M40$^{1.39}$, M82$^{2.53}$, M215$^{5.54}$, and M279$^{6.41}$ (superscripts indicate Ballesteros–Weinstein numbering[24]) (Fig. 1). M82$^{2.53}$, M215$^{5.54}$, and M279$^{6.41}$ assume distinctly different conformations between the crystal structures of $\beta_2$AR bound to an inverse agonist and $\beta_2$AR bound to a full agonist and a G protein, and the resonances from these residues of the $\beta_2$AR/4Met mutant exhibited large differences between each ligand-bound state[16, 18].

In the $^1$H-$^{13}$C HMQC spectra of the [$^2$H-9AA, $\alpha\beta\gamma$-$^2$H-, methyl-$^{13}$C-Met] $\beta_2$AR in rHDLs in the full agonist-bound state, the resonances from M36$^{1.35}$, M40$^{1.39}$, and M82$^{2.53}$ showed the same chemical shifts between the phosphorylated state and the unphosphorylated state. Interestingly, the $^1$H and $^{13}$C chemical shifts of the resonance from M215$^{5.54}$ were remarkably different

from those of the unphosphorylated $\beta_2$AR. Furthermore, the resonance from M279$^{6.41}$ was not observed, suggesting that this signal was broadened due to the exchange between the multiple conformations in the phosphorylated state (Fig. 3).

**Adhesion of the C-terminal region to the membrane surface.** For further investigation of the conformation of the phosphorylated $\beta_2$AR, the interaction between the C-terminal region and the membrane surface, which is composed of the intracellular side of the TM region of $\beta_2$AR and lipids, was examined by cross-saturation experiments[25]. As the amide resonances derived from the TM-proximal region could not be observed in the phosphorylated state, methyl resonances were utilized in the cross-saturation experiments. $\beta_2$AR in rHDLs, in which the C-terminal region of $\beta_2$AR and MSP1 are perdeuterated and the methyl groups of T360γ2 and I399δ1 of $\beta_2$AR are labeled with $^{13}$C, was irradiated with a frequency corresponding to the aromatic proton resonances, exclusively affecting $\beta_2$AR except for its C-terminal region, because no aromatic protons exist in the C-terminal region (Fig. 4a). The saturation in the aromatic protons of $\beta_2$AR is instantaneously transferred to all of the hydrogen atoms of the TM region of $\beta_2$AR and the lipids in rHDLs, which have high proton density, in a phenomenon known as the spin diffusion effect. Although the C-terminal region of $\beta_2$AR is not directly affected by the radiofrequency field, the saturation can be transferred from the aforementioned region to the C-terminal region, through the binding interface, in a phenomenon called the cross-saturation effect. The saturation transferred to the C-terminal region is limited to the interface, due to its low proton density. Therefore, the residues that are in close proximity to the membrane surface can be identified by observations of the signal intensity reductions of T360 and I399.

In the cross-saturation experiments using the phosphorylated $\beta_2$AR, the pT360 methyl resonances were remarkably affected by irradiation, whereas the methyl resonances from T360 to I399 were not significantly affected by irradiation in the experiments using unphosphorylated $\beta_2$AR (Fig. 4b, c). These results, together with the above-described phosphorylation-induced broadening of resonances from the amide groups in the TM-proximal region (Fig. 2b), suggested that the TM-proximal region of $\beta_2$AR adheres to the membrane surface in rHDLs upon phosphorylation.

**Conformation of the phosphorylated $\beta_2$AR bound to $\beta$-arrestin.** To investigate the effect of arrestin binding on the conformation of the TM region of the phosphorylated $\beta_2$AR, we observed the NMR resonances from the phosphorylated $\beta_2$AR mutant bound to $\beta$-arrestin 1. Surface plasmon resonance (SPR) analyses revealed that a $\beta_2$AR mutant binds to $\beta$-arrestin 1 in a phosphorylayion-dependent manner (Supplementary Fig. 4), in good agreement with the previously reported interaction between GPCRs and arrestins[26]. In the crystal structure of rhodopsin bound to visual arrestin, the residues corresponding to the methionine residues in $\beta_2$AR are >9 Å away from the arrestin atoms, suggesting that the resonances from methionine residues would be less affected by the direct interaction with $\beta$-arrestin 1, and would be good probes to investigate the conformational change of the TM region of $\beta_2$AR upon $\beta$-arrestin 1 binding. In the spectrum of the phosphorylated $\beta_2$AR mutant bound to both the full agonist and $\beta$-arrestin 1, the resonances from M215$^{5.54}$ and M279$^{6.41}$ exhibited different chemical shifts from those in the spectrum of the unphosphorylated $\beta_2$AR bound to the full agonist (Fig. 5). Comparison of the spectra in unphosphorylated, phosphorylated, and $\beta$-arrestin 1-bound state revealed that the M215$^{5.54}$ resonance in the phosphorylated state was observed at a

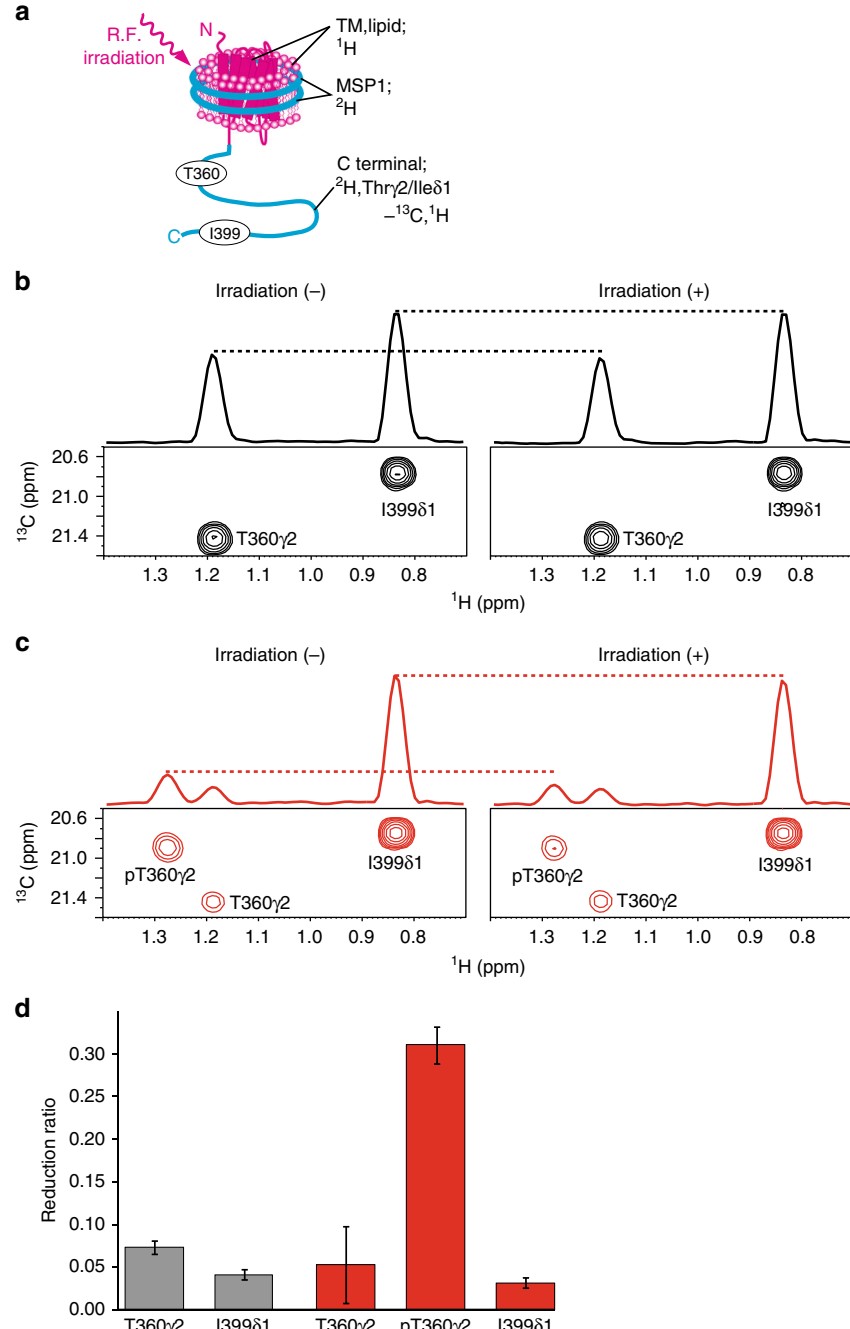

**Fig. 4** Interaction between the C-terminal region of β₂AR and the membrane surface examined by the methyl-directed cross-saturation experiments. **a** Schematic diagram of the methyl-directed cross-saturation experiment, using segmentally-labeled β₂AR. **b, c** Methyl-directed cross-saturation spectra of the C-terminal region of unphosphorylated β₂AR (**b**) and phosphorylated β₂AR (**c**). Spectra with and without irradiation are shown in the right and left panels, respectively. **d** Intensity reduction ratio of each resonance. Gray and red bars indicate the results in the unphosphorylated state and the phosphorylated state, respectively. The error bars represent the experimental errors, calculated from the root sum square of (noise level/signal intensity) in the two spectra, with and without irradiation

chemical shift between those in the unphosphorylated state and β-arrestin 1-bound state (Fig. 5).

## Discussion

The C-terminal regions of GPCRs are predicted to be intrinsically disordered, and the phosphorylated C-terminal regions and cytoplasmic cavity of the TM regions were assumed to interact with arrestins independently[27], although this is not necessarily an appropriate model for efficient complex formation between GPCRs and arrestins. In contrast, our cross-saturation

experiments using the segmentally labeled β₂AR in rHDLs revealed that the TM-proximal region of the C-terminal region adheres to the membrane surface, which is composed of the TM region or lipids in rHDLs (Fig. 4). The identified adhesion in this study results in the spatial proximity between the phosphorylated residues and the cytoplasmic cavity, which would be advantageous for arrestin to bind simultaneously to both the phosphorylated residues and the cytoplasmic cavity (Fig. 6).

Upon phosphorylation, the amide and methyl resonances derived from the TM-proximal region were broadened

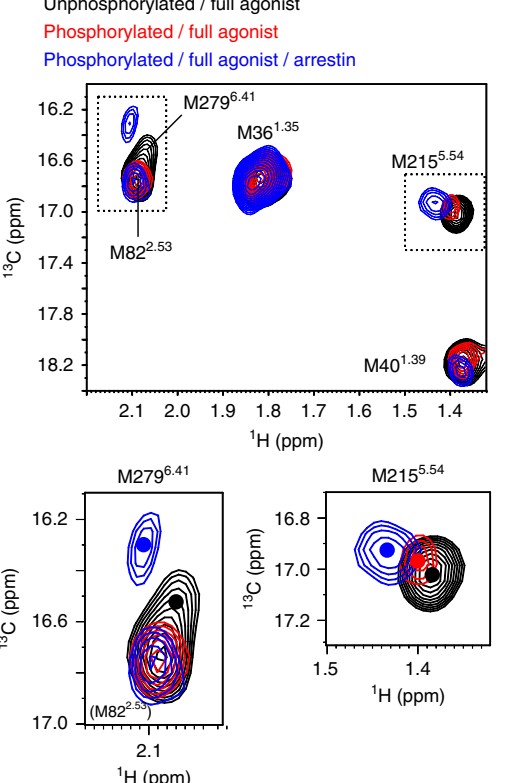

**Fig. 5** Conformation of the TM region of the phosphorylated $\beta_2$AR bound to $\beta$-arrestin. Overlay of the $^1$H-$^{13}$C HMQC spectra of [$^2$H-9AA, $\alpha\beta\gamma$-$^2$H, methyl-$^{13}$C-Met] $\beta_2$AR in rHDLs in the unphosphorylated state (black), phosphorylated state (red), and $\beta$-arrestin 1-bound state (blue). The centers of the resonances from M215$^{5.54}$ to M279$^{6.41}$ are indicated with dots. In the phosphorylated state, the resonance from M279$^{6.41}$ could not be observed

(Fig. 2b–d). The line broadening suggests that the phosphorylated TM-proximal region interconverts between multiple states, which are probably due to the negative-negative charge interaction between the phosphorylated TM-proximal region and lipids.

The cross-saturation experiments revealed that either the TM region of $\beta_2$AR or lipids adhere to the phosphorylated TM-proximal region (Fig. 4). Furthermore, we also observed the perturbation of the resonances from M215$^{5.54}$ and M279$^{6.41}$, which are located in the TM region, upon phosphorylation (Fig. 3). It is unlikely that the interaction between the phosphorylated C-terminal region and lipids induces the observed perturbation of the resonances from M215$^{5.54}$ and M279$^{6.41}$, because the methyl groups of M215$^{5.54}$ and M279$^{6.41}$ are buried in the TM helix bundle and >7 Å away from the lipid atoms. In addition, the cytoplasmic face of the TM region of $\beta_2$AR is positively charged (Supplementary Fig. 5a), which would complement the negatively charged phosphate groups incorporated within the C-terminal region. In the original crystal structure of rhodopsin from a native source[28], an electron density was observed in the intracellular core, and it is plausible that the density reflects the interaction between the phosphorylated C-terminal region and the TM region. Thus, it is most likely that the phosphorylated TM-proximal region interacts with the cytoplasmic face of the TM region by an electrostatic interaction, and the interaction alters the microenvironments around M215$^{5.54}$ and M279$^{6.41}$.

As in our previous NMR study of another GPCR[29], the chemical shifts of the methionine methyl groups in the TM region

sensitively reflected the conformational change of the TM helices. M215$^{5.54}$ and M279$^{6.41}$, which were perturbed upon phosphorylation in the NMR spectra of $\beta_2$AR, are located on the intracellular halves of TMs 5 and 6, respectively, and the $^1$H chemical shifts of M215$^{5.54}$ and M279$^{6.41}$ are affected by the ring current effects from F282$^{6.44}$ and Y219$^{5.58}$, respectively, which are sensitive to the spatial arrangement of the intracellular halves of TM5 and TM6, a possible arrestin binding site (Supplementary Fig. 6). These chemical shift perturbations of the methionine residues suggest that the intracellular halves of TM5 and TM6 become spatially rearranged upon phosphorylation. Interestingly, the M215$^{5.54}$ resonance shifted toward that in the $\beta$-arrestin 1-bound state upon phosphorylation (Fig. 5), suggesting that the phosphorylation-induced conformational change of TM5 and TM6, in addition to the adhesion of the phosphorylated C-terminal region to the membrane surface, would be prerequisite for $\beta$-arrestin recruitment (Fig. 6). Recent structural studies indicated that G-proteins or conformation-selective nanobodies modulate the conformation of the TM region from the intracellular side[30–34]. The proposed phosphorylation-induced conformational change of the TM region could be considered as a similar phenomenon. This is reasonable, since arrestin recognizes both the phosphorylated C-terminal region and the TM region.

The resonances from M215$^{5.54}$ of the phosphorylated $\beta_2$AR and the phosphorylated $\beta_2$AR–$\beta$-arrestin 1 complex exhibited a $^1$H downfield shift, in comparison with those of the unphosphorylated $\beta_2$AR. The observed $^1$H downfield shift would correspond to the inward movement of TM6, because F282$^{6.44}$, which defines the $^1$H chemical shift of M215$^{5.54}$, moves away from M215$^{5.54}$, with the inward movement of TM6 (Supplementary Fig. 6).

The crystal structures of GPCRs revealed that the degrees of the outward movement of TM6 vary among GPCRs in active states, from 3.5 to 14 Å (Supplementary Fig. 7). The proposed inward movement of TM6 upon phosphorylation should be included in the categories of these active state conformations. Particularly, in the structure of rhodopsin bound to arrestin, the degree of the outward movement of TM6 is smaller than that in the structure of $\beta_2$AR bound to G-protein, suggesting that arrestin recognizes a GPCR that exhibits a relatively small outward movement of TM6. The proposed inward movement of TM6 upon phosphorylation would thus be suitable for the phosphorylation-dependent arrestin recruitment.

To validate the proposed inward movement of TM6 upon phosphorylation, the capacity of the nanobody Nb6B9 was examined by SPR methods. Nb6B9 selectively recognizes the intracellular face of $\beta_2$AR in the active state, with a large outward movement of TM6[35]. As a result, the responses observed for Nb6B9 binding to the phosphorylated $\beta_2$AR were lower than those to the unphosphorylated $\beta_2$AR (Supplementary Fig. 8), which is consistent with the proposed phosphorylation-induced inward movement of TM6.

Phosphorylation of the TM-proximal region on $\beta_2$AR more effectively promotes the arrestin binding than that of the TM-distal region[36]. However, the mechanisms underlying the phosphorylation site-dependent arrestin recruitment are not understood. The phosphorylation-induced adhesion of the C-terminal region to the membrane surface and the conformational change of the TM region, which occur on the TM-proximal region but not on the TM-distal region, explain the phosphorylation site-dependent arrestin recruitment.

Different GRK subtypes reportedly phosphorylate different residues, and the patterns of the phosphorylated residues establish the phosphorylation barcode that dictates the conformation of bound $\beta$-arrestins and subsequent functions such as internalization and MAP kinase activation[37]. The previous

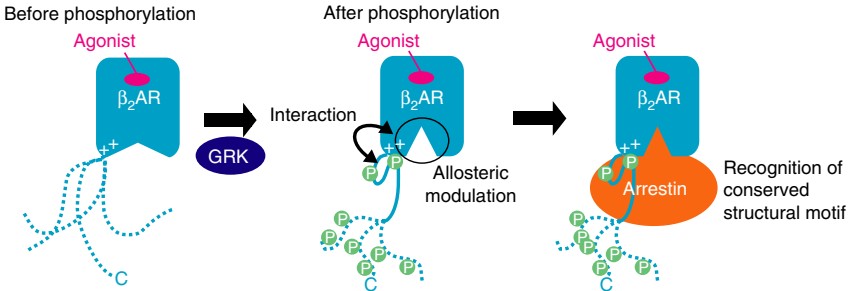

**Fig. 6** Role of the phosphorylation of the C-terminal region in signal transduction. In the unphosphorylated state, the C-terminal region of $\beta_2$AR is unstructured. Upon phosphorylation by GRKs, the TM-proximal region of the C-terminal region of $\beta_2$AR adheres to the cytoplasmic face of the TM region, and the intracellular halves of the TM helices become spatially rearranged, which preferentially activates arrestin-mediated signal transduction. The adhesion of the phosphorylated C-terminal region to the TM-region and the spatial rearrangement of the TM helices would generate the structural motif for $\beta$-arrestin binding

phosphorylation barcode theory solely focused on the position of the phosphorylated residues on the primary sequence. On the other hand, according to our NMR analyses, upon phosphorylation, the TM-proximal region adhered to the TM region and induced the conformational change of the TM helices, and that the TM-distal region was flexible even in the phosphorylated state. These results suggested that phosphorylation of the C-terminal region induced the conformational change dependent on the position of the phosphorylated residues and thus the conformation of the C-terminal region induced by the phosphorylation should be considered for understanding the regulation of the arrestin function.

The basic regions around the cytoplasmic faces of TM5 and TM6 are reportedly highly conserved among many GPCRs[38, 39], and the crystal structures of GPCRs other than $\beta_2$AR also contain positively charged clusters on the cytoplasmic face of the TM region (Supplementary Fig. 5b–d), although their role in receptor function is not well understood. It is plausible that the aforementioned conformation of the phosphorylated $\beta_2$AR, which is induced by the interaction between the phosphorylated residues and the positive charge clusters, is the general signature characteristic of GPCRs. $\beta$-arrestins can recognize dozens of activated and phosphorylated GPCRs, although the mechanism underlying the wide target specificity is not well understood. The conserved conformations of the phosphorylated GPCRs, characterized by the adhesion of the phosphorylated residues to the TM region and the inward movement of TM6 (Fig. 6), would be the structural motifs of GPCRs that enable $\beta$-arrestin to interact with dozens of GPCRs.

## Methods

**Statistics and general methods**. No statistical methods were used to predetermine sample size. The experiments were not randomized. The investigators were not blinded to allocation during experiment and outcome assessment.

**Reagents and buffers**. All reagents were from Nacalai Tesque, Inc., unless otherwise noted.

**Generation of recombinant baculovirus expressing TM-Int$_N$**. The DNA fragment encoding the hemagglutinin signal sequence, the FLAG-tag, previously reported E122W/N187E/C265A mutant of $\beta_2$AR (1–348, E122W, N187E, C265A), and the N fragment of the DnaE intein from *Nostoc punctiforme* was cloned into the pFastBac1 vector (Invitrogen). Primer sequences used to construct the plasmid are listed in Supplementary Table 2.

Sf9 cells (Invitrogen) were routinely maintained at 27 °C in Grace's supplemented medium (GIBCO) containing 10% fetal bovine serum (Biowest), 0.1% Pluronic F-68 (GIBCO), 50 IU mL$^{-1}$ penicillin, 50 μg mL$^{-1}$ streptomycin, and 0.125 μg mL$^{-1}$ amphotericin B. The recombinant baculovirus was generated and amplified with the Bac-to-Bac system (Invitrogen), according to the manufacturer's instructions.

**Expression and purification of TM-Int$_N$**. The *expres*SF+ cells (SF+ cells, Protein Science Corp.) were routinely maintained at 27 °C, in 100 mL Sf-900 II Serum free medium (GIBCO) in a 250 mL Erlenmeyer flask (Corning) on an orbital shaker (130 rpm). For the expression of TM-Int$_N$, the culture was expanded to 10 L in a 20 L Cellbag Bioreactor (GE Healthcare), using the Xuri cell expansion system W25 (GE Healthcare). When the cell density reached at about $2 \times 10^6$ cells mL$^{-1}$, 400 mL high-titer virus stocks, 1 μM alprenolol (SIGMA), and 14 μM E-64 (Peptide Institute Inc.) were added, and the culture was continued. Cells were harvested 48 h post-infection by centrifugation at $800 \times g$, and the resulting cell pellets were stored at −80 °C.

All of the following procedures were performed either on ice or in the cold room (4 °C). The cell pellet from 3 L of cell culture was suspended in 300 mL buffer A (50 mM Tris–HCl, pH 8.0, 100 mM NaCl, 0.1 mM tris(2-carboxyethyl) phosphine (TCEP), 1 mM ethylenediaminetetraacetic acid (EDTA), 1 mM 4-(2-aminoethyl) benzenesulfonyl fluoride hydrochloride (AEBSF), 20 μM leupeptin hemisulfate (Peptide Institute Inc.), 28 μM E-64 (Peptide Institute Inc.), 0.3 μM aprotinin (Wako Chemicals), 1 μM alprenolol (SIGMA)). The cells were disrupted by nitrogen cavitation (Parr Bomb) under 600 psi for 30 min. The cell lysate was centrifuged at $800 \times g$ for 10 min, and the resulting supernatant was centrifuged at $142,000 \times g$ for 40 min. The resulting pellet was washed twice in 80 mL buffer B (50 mM HEPES-NaOH, pH 7.2, 1 M NaCl, 20 mM KCl, 10 mM MgCl$_2$, 0.1 mM TCEP, 1 mM AEBSF, 20 μM leupeptin hemisulfate, 28 μM E-64) and centrifuged at $142,000 \times g$ for 40 min. The membrane pellet was resuspended in buffer C (20 mM HEPES-NaOH, pH 7.2, 150 mM NaCl, 20% (w/v) glycerol, 0.1 mM TCEP) supplemented with 1 μM alprenolol and was stored at −80 °C.

The membrane pellet from 3 L of cell culture was solubilized in 80 mL buffer C supplemented with 1% n-dodecyl-β-D maltopyranoside (DDM, Dojindo) for 3 h, and was then centrifuged at $142,000 \times g$ for 60 min. The supernatant was supplemented with 10 mM CaCl$_2$, and batch incubated with 8 mL ANTI-FLAG M1 Affinity Agarose Gel (SIGMA). The resin was washed with 100 mL buffer C, supplemented with 1 μM alprenolol, 0.1% DDM, and 3 mM CaCl$_2$. The protein was eluted with 20 mL buffer C, supplemented with 1 μM alprenolol, 0.1% DDM, 5 mM EDTA, and 0.2 mg mL$^{-1}$ DYKDDDDK peptide (Wako Chemicals).

The eluate from the ANTI-FLAG M1 Affinity Agarose Gel was further purified by chromatography on a Superose 6 10/300 GL column (GE Healthcare), equilibrated in buffer D (10 mM Tris–HCl, pH 7.5, 500 mM NaCl, 1 mM EDTA), supplemented with 0.1% DDM and 0.1 mM TCEP.

**Generation of the Int$_C$-Cterm plasmid**. The DNA fragment encoding a hexahistidine-tag, the C fragment of the DnaE intein from *Synechocystis species* PCC6803, and $\beta_2$AR (349–413, A349C, C378A, C406A) was cloned into the pRSF-1b vector (Novagen). The A349C/C378A/C406A mutations were introduced to improve the efficiency of the PTS reaction. We confirmed that the mutant retained the ability to be phosphorylated by GRK2 in a ligand-dependent manner and exhibited similar phosphorylation kinetics to those obtained in a previous study using wild-type $\beta_2$AR[40] (Supplementary Fig. 1). For the observation of the methyl resonances from T360γ2 to I399δ1, the T384A/T393A/S396A/S401A/S407A/T408A/S411A mutant was utilized for specifically investigating the effect of the phosphorylation of the TM-proximal region. Primer sequences used to construct the plasmids are listed in Supplementary Table 2.

**Expression and purification of Int$_C$-Cterm**. The *Escherichia coli* ER2566 strain (New England Biolabs), transformed with the Int$_C$-Cterm plasmid was cultured in 10 mL Luria Bertani (LB) medium containing 50 mg L$^{-1}$ kanamycin at 37 °C overnight. The cells were used to inoculate in 1 L of D$_2$O-based M9 medium, supplemented with 2 g L$^{-1}$ [$^2$H$_7$/$^{13}$C$_6$] glucose (Cambridge Isotope Laboratories, CIL) and 1 g L$^{-1}$ $^{15}$NH$_4$Cl (SI Science Co., Ltd.) as the sole carbon and nitrogen source, respectively. When the culture attained to an optical density at a wavelength of 600 nm (OD$_{600}$) of 0.8, 1 mM isopropyl-β-D-thiogalactopyranoside

(IPTG) was added, to induce protein expression. The culture was continued further at 37 °C for 6 h. For the selective $^{13}C^1H_3$ labeling of the threonine γ2 and isoleucine δ1 methyl groups, 50 mg L$^{-1}$ of [α,β-$^2H_2$, γ2-$^{13}C^1H_3$] threonine (NMR Bio), 100 mg L$^{-1}$ of [$^2H_2$] glycine (NMR Bio), and 50 mg L$^{-1}$ of [methyl-$^{13}$C, 3,3-$^2$H] ketobutyric acid (CIL) were supplemented into the medium, 1 h before the addition of IPTG. The cells were harvested by centrifugation at 5000 × $g$ for 15 min, and resulting cell pellets were stored at −80 °C.

The cells from a 1 L culture were suspended in 60 mL buffer E (50 mM Tris–HCl, pH 8.0, 300 mM NaCl), supplemented with 2 mM dithiothreitol (DTT). The cells were disrupted by sonication, and the cell lysate was centrifuged at 100,000 × $g$ for 1 h. The supernatant was applied to a 1 mL HisTrap HP column (GE Healthcare), equilibrated in buffer E. The protein was eluted by a linear concentration gradient of imidazole, from 25 mM to 190 mM. The eluate was further purified by reverse-phase high performance liquid chromatography (HPLC), using a YMC-Pack ODS AM-323 column (YMC), connected to Prominence HPLC system (SHIMADZU). The eluate was lyophilized, dissolved in H$_2$O, pH adjusted to 7.5, and lyophilized again.

**Ligation of TM-Int$_N$ and Int$_C$-Cterm**. Purified TM-Int$_N$ was concentrated, using a centrifugal filter device (AmiconUltra-4, 30 kDa molecular weight cut off, Millipore). Lyophilized Int$_C$-Cterm was dissolved in a concentrated TM-Int$_N$ solution, and incubated at 25 °C in the presence of 0.5 mM TCEP for 6 h. The ligation reaction was monitored by SDS–PAGE.

**Reconstitution of ligated β$_2$AR into rHDLs**. The *Escherichia coli* BL21 (DE3) (Stratagene), transformed with the plasmid encoding His-tagged MSP1[15], was cultured at 37 °C in Terrific Bloth (TB) media containing 100 mg L$^{-1}$ of ampicillin. When the culture attained an OD$_{600}$ of 2.0, 1 mM of IPTG was added to induce protein expression. The culture was continued further at 37 °C for 3 h. The cells were harvested by centrifugation at 5000 × $g$ for 15 min, and the resulting cell pellets were stored at −80 °C.

The cells were suspended in buffer F (50 mM Tris–HCl (pH7.5), 300 mM NaCl, 100 mM KCl) and disrupted by sonication. The cell lysate was centrifuged at 20,000 × $g$ for 30 min, and the pellet was suspended in buffer F, supplemented in 1% Triton X-100. The suspension was centrifuged at 100,000 × $g$ for 1 h, and the supernatant was applied to HIS-Select resin (SIGMA). The resin was washed with buffer F supplemented in 1 % Triton X-100, buffer F supplemented with 50 mM Na-cholate, and buffer F supplemented with 20 mM imidazole. The His-tagged MSP1 was eluted with buffer F supplemented with 200 mM imidazole. Subsequently, TEV protease was added to the eluate, which was dialyzed against buffer G (50 mM Tris–HCl, 10 mM NaCl, 0.5 mM EDTA) for 1 day. The sample was then dialyzed against buffer C, and passed through the HIS-Select resin to obtain the MSP1 without His-tag.

For the preparation of the biotinylated rHDL, C-terminally AVI-tagged MSP1 was co-expressed with a biotin ligase BirA in *Escherichia coli* AVB 101 (Avidity).

A mixture of 1-palmitoyl-2-oleoyl-phosphatidylcholine (POPC, Avanti Polar Lipids) and 1-palmitoyl-2-oleoyl-phosphatidylglycerol (POPG, Avanti Polar Lipids) in chloroform was prepared at a molar ratio of 3:2. The solvent was evaporated under a nitrogen atmosphere and dried in vacuo to form a lipid film. The film was solubilized in buffer H (20 mM HEPES-NaOH, pH 7.2, 150 mM NaCl, 0.1 mM TCEP), supplemented with 100 mM sodium cholate, for a final lipid concentration of 50 mM.

All of the following procedures were performed either on ice or in the cold room (4 °C) unless otherwise stated. The prepared MSP1 and the lipid solution were added to the ligation reaction mixture at final concentrations of 0.1 mM and 5 mM, respectively. The mixture was incubated on ice for 3 h. Afterwards, 80% (w/v) of Bio-Beads SM-2 (Bio-Rad) was added to the mixture, which was incubated in the cold room overnight with gentle mixing. The supernatant was supplemented with 10 mM CaCl$_2$, and applied to a 1 mL ANTI-FLAG M1 Affinity Agarose Gel column, equilibrated in buffer H supplemented with 3 mM CaCl$_2$. The resin was washed with 5 mL buffer H supplemented with 3 mM CaCl$_2$. The β$_2$AR in rHDLs were eluted with buffer H, supplemented with 5 mM EDTA and 0.2 mg mL$^{-1}$ DYKDDDDK peptide.

For the preparation of the phosphorylated β$_2$AR in rHDL, the elution buffer was exchanged to buffer I (20 mM Tris–HCl, pH 7.5, 10 mM MgCl$_2$, 0.1 mM TCEP), using a PD-10 desalting column (GE Healthcare). Purified GRK2, ATP, and formoterol were added at final concentrations of 1.5 μM, 1 mM, and 100 μM, respectively. The reaction mixture was incubated at 30 °C for 1.5 h, and was analyzed by SDS–PAGE with Pro-Q® Diamond and SYPRO®-Ruby (Molecular Probes) staining. Gel images were obtained with a Typhoon FLA 9000 imager (GE Healthcare).

The eluate from the ANTI FLAG M1 Affinity Agarose Gel or the phosphorylated mixture was further purified on a Superdex 200 10/300 GL increase column (GE Healthcare) equilibrated in buffer H. The fractions corresponding to a Stokes diameter of ~ 11 nm were collected, supplemented with 20 μM formoterol, and concentrated using a centrifugal filter device (Amicon Ultra-4 10 kDa of molecular weight cut-off, Millipore), while exchanging the buffer to buffer J (20 mM PIPES-NaOH, pH 6.8, 2 mM EDTA, 0.1 mM TCEP, H$_2$O/D$_2$O = 90/10).

**Preparation of the β$_2$AR 4Met mutant embedded in rHDLs**. The DNA fragment encoding the hemagglutinin signal sequence, the FLAG-tag, β$_2$AR (2–413) with the 4Met mutation, and a decahistidine-tag, was cloned into the pFastBac1 vector (Invitrogen). The generation of the recombinant baculovirus was performed as described above for TM-Int$_N$. For the expression of [$^2$H-9AA, αβγ-$^2$H, methyl-$^{13}$C-Met] β$_2$AR, SF+ cells were suspended in ESF921 ΔAA media (Expression Systems), supplemented with 3 g L$^{-1}$ aspartic acid sodium salt, 2 g L$^{-1}$ glutamic acid sodium salt, 1.5 g L$^{-1}$ glutamine, 1.5 g L$^{-1}$ asparagine, 1.0 g L$^{-1}$ lysine hydrochloride, 1.0 g L$^{-1}$ arginine hydrochloride, 0.5 g L$^{-1}$ glycine, 0.5 g L$^{-1}$ serine, 0.5 g L$^{-1}$ proline, 0.5 g L$^{-1}$ hydroxyproline, 0.3 g L$^{-1}$ histidine, 0.1 g L$^{-1}$ [ring-$^2$H]-tryptophan (CIL), 0.05 g L$^{-1}$ [αβγ-2H, methyl-$^{13}$C]-methionine (CIL), 0.27 g L$^{-1}$ [β-$^2$H]-DL-cysteine (CIL), and 0.1 g L$^{-1}$ [$^2$H]-DL-tyrosine, at about 2 × 10$^6$ cells mL$^{-1}$. The cells were cultured with high-titer virus stock. At 20 h post infection, 1.5 g L$^{-1}$ [$^2$H]-algal amino-acid mixture (CIL), 0.1 g L$^{-1}$ [αβγ-$^2$H, methyl-$^{13}$C]-methionine, 0.05 g L$^{-1}$ [$^2$H]-phenylalanine (CIL), 0.05 g L$^{-1}$ [$^2$H]-valine (CIL), 1 g L$^{-1}$ [$^2$H]-alanine, and 0.01 g L$^{-1}$ β-chloro-L-alanine hydrochloride, were added to the culture. Cells were collected at 48 h post infection by centrifugation at 800 × $g$, and resulting pellet was stored at −80 °C.

The preparation of membrane and solubilization by DDM was performed as described for the purification of TM-Int$_N$. The supernatant was batch incubated with TALON resin for overnight. The resin was washed with buffer C supplemented with 0.1% DDM and 20 mM imidazole. The protein was eluted with buffer C supplemented with 0.1% DDM and 200 mM imidazole. The eluate was batch incubated for 4 h with Affi-gel 10 (Bio-Rad), on which 5 mg mL$^{-1}$ alprenolol-cysteamine was immobilized. The resin was washed with buffer C, supplemented with 0.1% DDM. β$_2$AR was eluted with buffer C supplemented with 0.1 % DDM and 1 mM formoterol.

For the analysis of β$_2$AR bound to β-arrestin 1, the amino acid sequence of the β$_2$AR C-terminal region (residues 342–413) was substituted with that of the human arginine vasopressin V2 receptor C-terminal region (residues 343–371), to achieve the high-affinity interaction with β-arrestin 1[41]. For the purification of this mutant, the His-tag purification steps using TALON resin (Clontech) were replaced by the FLAG-tag purification steps, using ANTI-FLAG M1 Affinity Agarose Gel (SIGMA) because this mutant lacked the C-terminal His-tag. The FLAG-tag purification was performed as described for the purification of TM-Int$_N$.

**Generation of the β-arrestin 1 plasmid**. The DNA fragment encoding the hexahistidine-tag, the TEV-protease cleavage site (ENLYFQG), and the human β-arrestin 1 cysteine-less mutant[42] was cloned into the pTrcHisB vector (Thermo Fisher Scientific). Primer sequences used to construct the plasmid are listed in Supplementary Table 2.

**Expression and purification of β-arrestin 1**. The *Escherichia coli* BL21 (DE3) codon plus RP strain (Stratagene), transformed with the β-arrestin 1 plasmid was cultured in 10 mL Luria Bertani (LB) medium containing 100 mg L$^{-1}$ ampicillin at 37 °C overnight. The overnight culture was used to inoculated in 1 L of D$_2$O-based M9 medium, supplemented with 2 g L$^{-1}$ [$^2$H$_7$] glucose and 1 g L$^{-1}$ $^{15}$NH$_4$Cl as the sole carbon and nitrogen sources, respectively. When the culture attained an OD$_{600}$ of 0.8, 0.3 mM isopropyl-β-D-thiogalactopyranoside (IPTG) was added to induce protein expression. The culture was continued further at 30 °C for 20 h. For the selective $^{13}C^1H_3$ labeling of the isoleucine δ1 methyl group, 50 mg L$^{-1}$ of [methyl-$^{13}$C, 3,3-$^2$H] ketobutyric acid were supplemented into the medium 1 h before the IPTG addition. The cells were harvested by centrifugation at 5000 × $g$ for 15 min, and the resulting cell pellets were stored at −80 °C.

The cells from a 1 L culture were suspended in 60 mL buffer K (20 mM Tris-HCl, pH 8.0, 200 mM NaCl) supplemented with Protease Inhibitor Cocktail. The Cells were disrupted by sonication, and the cell lysate was centrifuged at 100,000 × $g$ for 1 h. The supernatant was applied to a 1 mL HisTrap HP column (GE Healthcare), equilibrated in buffer K. The protein was eluted by a linear concentration gradient of imidazole, from 25 mM to 175 mM. Subsequently, 1 mg of TEV protease was added to the eluate, which was dialyzed against buffer L (20 mM Tris–HCl, pH 8.0, 2 mM EDTA) supplemented with 150 mM NaCl, for 16–18 h. The samples were applied to a 1 mL HiTrap Heparin HP (GE Healthcare), equilibrated with buffer L. The protein was eluted by a linear concentration gradient of NaCl, from 200 mM to 700 mM. The eluate was concentrated using a centrifugal filter device (Amicon Ultra-4 30 kDa molecular weight cut-off, Millipore), and further purified on a Superdex 75 10/300 GL column (GE Healthcare), equilibrated in buffer L supplemented with 300 mM NaCl. The eluate was concentrated using a centrifugal filter device (Amicon Ultra-4 30 kDa molecular weight cut-off, Millipore), while exchanging the buffer to buffer F prepared in H$_2$O/D$_2$O = 1/99.

**Preparation of Nb6B9**. The DNA fragment encoding the C-terminally H$_8$-tagged Nb6B9[35] was cloned into the pET26b vector (Novagen) by using restriction enzymes NdeI and EcoRI (TAKARA BIO). The resulting sequence to be translated contains additional two residues (Met-Asp) at the N terminus and a lysine residue just upstream of the C-terminal H$_8$-tag.

The *Escherichia coli* SHuffle T7 Express[43] (NEW ENGLAND Biolabs) transformed with the Nb6B9 plasmid was cultured at 30 °C in LB media containing

30 mg L$^{-1}$ of kanamycin. When the culture attained an OD$_{600}$ of 0.8, 0.4 mM of IPTG was added to induce protein expression. The culture was continued further at 16 °C for 16 h. The cells were collected by centrifugation at 5000 × $g$ for 15 min, and the resulting cell pellets were stored at −80 °C.

The cells were suspended in buffer M (50 mM Tris–HCl (pH7.5), 150 mM NaCl, 1 mM phenylmethylsulfonyl fluoride) and disrupted by sonication. The cell lysate was centrifuged at 27,000 × $g$ for 30 min, and the supernatant was purified by the His-tag affinity chromatography using TALON resin (Clontech). The fractions containing Nb6B9 were pooled and further purified by size exclusion chromatography using Superdex 75 10/300 GL column (GE lifesciences) with buffer N (20 mM Hepes-NaOH (pH 6.5), 300 mM NaCl).

**Surface plasmon resonance experiments.** The β-arrestin 1 and Nb6B9 binding activities of unphosphorylated or phosphorylated β$_2$AR mutants were analyzed using Biacore T100 or Biacore T200 instrument (GE Healthcare). The β$_2$AR mutants embedded in biotinylated rHDLs were captured on the flow cells of the Series S Sensor Chip SA (GE Healthcare) via the interaction between biotin and streptavidin. The binding assays were carried out in buffer H, supplemented with 1 µM formoterol, at a flow rate of 30 µL min$^{-1}$. The obtained sensorgrams were processed using Biacore T-200 Evaluation software (GE Healthcare).

**NMR experiments.** All spectra were recorded with a Bruker AVANCE III 800 or Bruker AVANCE III 950 spectrometer equipped with a cryogenic probe, processed by Topspin 3.1 (Bruker), and analyzed by Sparky[44]. The $^1$H chemical shifts were referenced to the methyl protons of 3-(trimethylsilyl)-1-propanesulfonic acid sodium salt, and the $^{13}$C and $^{15}$N chemical shifts were referenced indirectly.

$^1$H-$^{15}$N HSQC spectra with water flip-back and WATERGATE were recorded for 10–20 µM {Cterm-[$^2$H,$^{13}$C,$^{15}$N]} β$_2$AR in rHDLs in buffer J, supplemented with 20 µM formoterol. The spectral widths were set to 16 and 22 ppm for the $^1$H and $^{15}$N dimension, respectively, and the inter-scan delays were set to 1 s. In total, 512 × 128 complex points were recorded, and 256 scans/FID gave rise to an acquisition time of 20 h for each spectrum.

TROSY-HNCA and TROSY-HN(CO)CA were recorded for ~ 20 µM {Cterm-[$^2$H,$^{13}$C,$^{15}$N]} β$_2$AR in rHDLs in buffer J, supplemented with 20 µM formoterol. The spectral widths were set to 16, 22, and 24 ppm for the $^1$H, $^{13}$C, and $^{15}$N dimensions, respectively, and the inter-scan delays were set to 1 s. In total, 1,024 × 30 × 32 complex points were recorded, and 24 scans/FID gave rise to an acquisition time of 36 h for each spectrum.

$^1$H-$^{13}$C HMQC spectra with echo/anti-echo gradient coherence selection were recorded for ~ 10 µM {Cterm- [$^2$H, Thrγ2/Ileδ1-$^{13}$C$^1$H$_3$]} β$_2$AR in rHDLs in buffer J, prepared in H$_2$O/D$_2$O = 1/99 and supplemented with 20 µM formoterol. The spectral widths were set to 16 and 8 ppm for the $^1$H and $^{13}$C dimension, respectively, and the inter-scan delays were set to 1 s. In total, 512 × 128 complex points were recorded, and 256 scans/FID gave rise to an acquisition time of 20 h for each spectrum.

Methyl-utilizing intramolecular CS spectra were recorded for ~ 10 µM {Cterm-[$^2$H, Thrγ2/Ileδ1-$^{13}$C$^1$H$_3$]} β$_2$AR in rHDLs in buffer J, prepared in H$_2$O/D$_2$O = 1/99 and supplemented with 20 µM formoterol. The irradiation frequency was set at 5.0 ppm, and the maximum radiofrequency amplitude was 0.21 kHz for WURST-20 (the adiabatic factor Q$_0$ = 1). The irradiation times and the additional relaxation time were set to 0.5 and 1.5 s, respectively.

$^1$H-$^{13}$C HMQC spectra were recorded for ~ 20 µM [$^2$H-9AA, αβγ-$^2$H-methyl-$^{13}$C-Met] β$_2$AR 4Met mutant embedded in rHDLs. The spectral widths were set to 16 and 32 ppm for the $^1$H and $^{13}$C dimension, respectively, and the inter-scan delays were set to 1 s. In total, 512 × 128 complex points were recorded, and 256 scans/FID gave rise to an acquisition time of 20 h for each spectrum.

**Data availability.** The data that support the findings of this study are available from the corresponding author upon reasonable request.

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

## Acknowledgements

This work was supported by the development of core technologies for innovative drug development based upon IT, from the Japan Agency for Medical Research and Development, AMED, and by The Ministry of Education, Culture, Sports, Science and Technology (MEXT)/Japan Society for the Promotion of Science (JSPS) KAKENHI Grant Numbers JP15J12409, JP15K18843, JP16H01353, JP16H01531, JP17H04999, JP17H06097. Y.S. and M.N. are JSPS fellows. H.I. was supported by the Academy of Finland (137995, 1277335) and Biocenter Finland. The NMR experiments were partly performed at Yokohama City University of NMR Platform supported by the MEXT, Japan.

## Author contributions

Y.S., M.N., Y.K., T.U., H.I., and I.S. designed the research. Y.S., M.N., S.I., K.N., and T.M. performed the research. Y.S., M.N., Y.K., T.U., H.I., and I.S. analyzed the data. Y.S., M.N., Y.K., S.I., T.U., H.I., and I.S. wrote the manuscript.

## Additional information

**Competing interests:** The authors declare no competing financial interests.

