## [Peer Review File · Nature Communications]

Reviewers' comments:

Reviewer #1 (Remarks to the Author):

The manuscript by Shimada and coworkers describes high-resolution solution NMR measurements on the β 2-adrenergic receptor in an unphosphorylated state and in a phosphorylated form in the absence and presence of arrestin.

Functional studies on GPCRs have shown that they can exist in a number of specific conformations that are stabilized by different ligands and G proteins according to cell type and conditions. While crystallography can capture distinct, highly populated low energy states, it provides only limited information about dynamic aspects of protein structure, and is not well-suited for high-resolution measurements of receptors in membrane environments. NMR spectroscopy, on the other hand, is able to address questions of structure and dynamics of GPCRs in both detergent micelles and membrane bilayers.

Nevertheless, NMR approaches to GPCRs are not simple. As with both structural and functional studies, they require expression and purification of functional receptor from mammalian or insect cells. However, in addition, for this study the authors have undertaken segmental labeling of the C-terminus to target the arrestin-binding region, and have reconstituted the receptor in HDL particles. The data they have obtained are of high quality and the comparative approach (i.e. comparing samples in three different states) not only provides a check on the ability of their reconstituted receptors to respond to phosphorylation and arrestin binding, but also provides insights into the mechanism of arrestin binding. In terms of the methodology alone, this work advances the field and merits publication.

The advantage of the NMR approach is that it yields sequence specific information on structure and dynamics. In the absence of a full NMR structure (which is not the goal here), the challenge is drawing conclusions from the data that the non-expert can build on for further studies. There are three major conclusions that are made in the Abstract (and manuscript) and I feel none of these three conclusions as written are fully supported by the data.

1. In the Abstract, the authors conclude that "Our analyses revealed that, firstly, the phosphorylated C-terminal region adheres to the cytoplasmic face of the transmembrane region, which is positively charged." This conclusion is based on cross saturation experiments that rely on proton spin diffusion. In the Results section they correctly indicate that this means that this region either interacts with the TM helices (or loops) or the lipids. However, in both the abstract and discussion the authors conclude it interacts with the TM helices and discard the alternative explanation that the influence on the TM-proximal region of the C-terminus exhibits larger changes than the TM-distal region because the TM-proximal region is closer to the membrane surface (either TM helices or lipids).

(i.e. I understand that the authors believe that the interaction with the TM helices makes more sense in terms of the charge complementary. However, just because this makes sense does not mean it is correct.)

The wording used in the Results or Supporting Information is less misleading than in the Abstract.

Results (page 11). "These results, together with the above-described phosphorylation-induced broadening of resonances from the amide groups in the TM-proximal region (Fig. 2b), suggested that the TM-proximal region of β 2AR adheres to the TM region or lipids in rHDLs upon phosphorylation."

Supporting Info (page 3). "...the cross-saturation experiments suggested that the phosphorylated

residues in the TM-proximal region, which is required for the arrestin recruitment³, interact with the transmembrane region.”

2. In the Abstract, the authors conclude that they “found that these structures induced by the phosphorylation are preferable for the interaction of β 2AR with β -arrestin.” The direct implication is that the structures induced are preferable relative to the non-phosphorylated forms with β -arrestin. However, the measurements in Figure 5 that describe the arrestin interactions are only for β -arrestin in the phosphorylated state. Measurements of the non-phosphorylated receptor + β -arrestin are not reported. The authors cannot rule out that a large change in chemical shifts of the Met residues would be observed upon addition of β -arrestin to the non-phosphorylated receptor (containing full agonist) without running the experiment. It may be that the full agonist allows an induced fit of β -arrestin.

3. In the Abstract, the authors conclude that “Our analyses revealed that... secondly, the interaction induces the spatial rearrangement of the transmembrane helices toward the conformation that corresponds to the β -arrestin-bound state.” This conclusion is based on the observation of two Met side chain methyl group resonances (Met5.54 and Met6.41). Actually, the conclusion is based on only one Met side chain methyl chemical shift (Met5.54) since the Met6.41 chemical shift in the phosphorylated receptor is not observed. (It might be that this resonance overlaps with that of Met2.53 or is shifted in a different direction.) The single Met5.54 chemical shift ($\Delta^{13}C \sim 0.1$ ppm) does not directly report on the spatial rearrangement of the transmembrane helices. For the non-NMR reader, this conclusion would be very misleading.

The issue I have with the manner in which the authors state their conclusions continues in the Discussion section. In a general way, the use of the term “Therefore” in my opinion is the final statement in a logical argument (either through deductive or inductive reasoning). In a deductive argument, if one accepts the premises then the conclusion must be true (i.e. a simple form would be “If A, then B. One observes A. Therefore, B.”) In an inductive argument, the conclusion is not necessarily true. An example is that one concludes after observing ravens for several days that “All ravens are black”. It may well be the case that there is a raven that is white that was missed.

Paragraph 2 of the Discussion is an example (the first of six instances of using the term “Therefore” in the Discussion).

The cytoplasmic face of the TM region of β 2AR is positively charged (Supplementary Fig. 5a), whereas the lipids in rHDLs and the phosphate groups attached to the C-terminal region are negatively charged. Therefore, the observed cross-saturation effect is likely to correspond to the electrostatic interaction with the cytoplasmic face of the TM region.

The cross saturation effect may well correspond to a positive-negative interaction. However, an explanation that would cover BOTH the broadening of the amide resonances upon phosphorylation and the cross saturation effect would be that the negative-negative interaction (C-terminus and phosphate groups) leads to an ensemble of conformations (i.e. there is no single defined state as there might be with complementary interactions) and hence broadening. The cross-saturation of the TM proximal residues (vs. the TM-distal) residues is just because they are closer.

One might say that the positive-negative complementarity between the phosphate groups and the cytoplasmic surface of the receptor suggests that these regions associate and would be consistent with the cross saturation results.

In short, I think the approach taken and the data presented are very strong and merit publication in Nature Communications. The authors simply need to clearly distinguish (particularly in the Title and Abstract) what can directly be concluded from the data and what one might speculate based on these

conclusions.

Reviewer #2 (Remarks to the Author):

This manuscript describes an interesting study on the effects of phosphorylation on conformation of the beta2 adrenergic receptor. The authors utilize several NMR spectroscopy-based approaches to study the role that phosphorylation plays in managing receptor conformation and arrestin binding. In addition to methionine labeling, the authors exploit intein-based fusion technology to replace the receptor's C-terminus with an isotope-labeled carboxy terminus produced in bacteria. Upon reconstitution in a lipid bilayer and in vitro phosphorylation, the authors were able to discern specific conformational changes in the TM region as well as in the C-terminus following phosphate incorporation and arrestin binding. These are novel approaches applied to G protein-coupled receptors and novel findings.

Although this is a very interesting study there are several issues that the authors need to address in their manuscript, preferably either through additional experimentation or minimally through detailed, extensive discussion.

Comments:

The cross saturation results are quite interesting. Some reference to the highly conserved basic regions of GPCRs around the cytosolic faces of TM5 and TM6 should be made. Hundreds of reports focusing on these basic regions have been published throughout the years but a strong consensus of their role in receptor function has not been met. Structural evidence, particularly the GPCR-G protein complex structures have provided little role for these regions in G protein coupling. The interaction of the phosphorylated C-terminal tail, as suggested in this current study, could finally provide some rationale for their conservation. Also, it should be noted that the original crystal structure of rhodopsin (Palczewski et al 2000) reported a density in the intracellular core (where the G protein and arrestin binds) that was modelled as the receptor's C-terminus. Granted this was a crystal structure however the receptor was obtained from native sources (not recombinant) and was not treated with a phosphatase. Limited, or partial phosphorylation the rhodopsin C-terminus can not be ruled-out in this preparation.

The reference to and discussion of the beta2AR-arrestin study by electron microscopy (and functional analyses) should be tempered in that the receptor used in the study was a C-terminal fusion with the V2R receptor. The V2R is a class B set of receptors that displays sustained interactions with arrestin. Its interaction with arrestin will be different than that with class A receptors like the beta2AR. This is an artificial system that was used to stabilize a complex between arrestin and the beta2AR. This is not the system used in this current study and therefore comparisons should not be made. Note that the class designation should not be confused with the Family A/B/C/D/E/F GPCR nomenclature and only refers to the interaction with arrestin. The reference to the beta2AR-arrestin structure was made during the discussion of the potential contribution of the phospholipid bilayer under which these current experiments were performed. Indeed, it is likely that the lipid environment plays a critical role, not tested previously in this manner, but the comparison with the previously reported EM work should not be made.

The phosphorylation-dependent movement of M215 is quite interesting. This, coupled with the chemical shift observed on M279, suggests that phosphorylation influences the conformation of the TM regions of the receptor. Although the authors provide a detailed discussion of the phenomena the authors do not address the mechanistic relevance. Inward movement of TM6, as observed following

phosphorylation, would normally be associated with the inactive conformation of the receptor. In contrast, as the authors know, is associated with the outward movement of TM6, a form that can be stabilized by G proteins, conformation-selective antibodies, and arrestins. Some discussion of the discrepancy must be made. It would be even better if the authors provided biochemical or biophysical data that tested the capacity of the G proteins or conformation-selective antibodies to couple to the phosphorylated receptor. For example, does agonist binding change in a phosphorylation-dependent manner in this preparation? Inward movement of TM6, or stabilization of the inactive conformation, should decrease agonist binding affinity. Do G proteins or conformation-selective antibodies (nanobodies) influence the chemical shifts observed on the phosphorylated receptor? The nanobodies (eg Nb80) in this case would be quite powerful as they should be able to bind the phosphorylated receptor.

In the discussion section (lines 258-268) the authors state that the "results clearly suggested that phosphorylation of the C-terminal regions induced the conformational changes dependent on the position of the phosphorylated residues", in reference to the phosphorylation bar code hypothesis. The term 'clearly' is quite strong here based on the data provided here.

Finally, several studies have been reported on the structural analysis of agonist-bound GPCRs, using crystallography, EPR and NMR spectroscopy. In each case it was difficult to observe, from a population perspective, stabilization of the active conformation with an agonist alone (ie. without a G protein or conformation-selective antibody). The fact that GRK-mediated phosphorylation occurs in an active receptor-dependent manner, suggests that the result of GRK interaction, phosphorylation, should also maintain the receptor in an active conformation. This would make some sense since it would phosphorylation enhances arrestin recruitment both the C-terminus and to the receptor core. Some discussion of these properties in the context of the experimental results of this study should be made.

Minor: the cartoon of the phosphorylated residues interacting with the basic region (of TM5 and TM6) should be modified. It is a little misleading to suggest that the residues are as deep into the cytosolic core of the receptor where the G proteins and arrestin bind.

Responses to Reviewers

Reviewer #1

Comment 1-1

The manuscript by Shimada and coworkers describes high-resolution solution NMR measurements on the β 2-adrenergic receptor in an unphosphorylated state and in a phosphorylated form in the absence and presence of arrestin.

Functional studies on GPCRs have shown that they can exist in a number of specific conformations that are stabilized by different ligands and G proteins according to cell type and conditions. While crystallography can capture distinct, highly populated low energy states, it provides only limited information about dynamic aspects of protein structure, and is not well-suited for high-resolution measurements of receptors in membrane environments. NMR spectroscopy, on the other hand, is able to address questions of structure and dynamics of GPCRs in both detergent micelles and membrane bilayers.

Nevertheless, NMR approaches to GPCRs are not simple. As with both structural and functional studies, they require expression and purification of functional receptor from mammalian or insect cells. However, in addition, for this study the authors have undertaken segmental labeling of the C-terminus to target the arrestin-binding region, and have reconstituted the receptor in HDL particles. The data they have obtained are of high quality and the comparative approach (i.e. comparing samples in three different states) not only provides a check on the ability of their reconstituted receptors to respond to phosphorylation and arrestin binding, but also provides insights into the mechanism of arrestin binding. In terms of the methodology alone, this work advances the field and merits publication.

The advantage of the NMR approach is that it yields sequence specific information on structure and dynamics. In the absence of a full NMR structure (which is not the goal here), the challenge is drawing conclusions from the data that the non-expert can build on for further studies. There are three major conclusions that are made in the Abstract (and manuscript) and I feel none of these three conclusions as written are fully supported by the data.

I appreciate the reviewer's comment that the three conclusions described in the abstract should be supported by the experimental data. We explain the revisions about the three conclusions in the following responses to Comments 1-2, 1-3, and 1-4.

Comment 1-2

1. In the Abstract, the authors conclude that “Our analyses revealed that, firstly, the phosphorylated C-terminal region adheres to the cytoplasmic face of the transmembrane region, which is positively charged.” This conclusion is based on cross saturation experiments that rely on proton spin diffusion. In the Results section they correctly indicate that this means that this region either interacts with the TM helices (or loops) or the lipids. However, in both the abstract and discussion the authors conclude it interacts with the TM helices and discard the alternative explanation that the influence on the TM-proximal region of the C-terminus exhibits larger changes than the TM-distal region because the TM-proximal region is closer to the membrane surface (either TM helices or lipids).

(i.e. I understand that the authors believe that the interaction with the TM helices makes more sense in terms of the charge complementary. However, just because this makes sense does not mean it is correct.)

The wording used in the Results or Supporting Information is less misleading than in the Abstract.

Results (page 11). “These results, together with the above-described phosphorylation-induced broadening of resonances from the amide groups in the TM-proximal region (Fig. 2b), suggested that the TM-proximal region of β 2AR adheres to the TM region or lipids in rHDLs upon phosphorylation.”

Supporting Info (page 3). “...the cross-saturation experiments suggested that the phosphorylated residues in the TM-proximal region, which is required for the arrestin recruitment³, interact with the transmembrane region.”

I appreciate the reviewer's comment that the possibility that the cross-saturation reflects the interaction with lipids is ruled out in the Abstract and Discussion sections. In the revised manuscript, we changed the descriptions in the Abstract and Discussion sections, as follows.

In the Abstract section, we added the following sentence to describe the conclusion directly derived from the cross-saturation experiments (page 2, lines 7-8):

“Our analyses revealed that the phosphorylated C-terminal region adheres to either the

intracellular side of the transmembrane region or lipids”

In the Discussion section, we added the descriptions that the interaction between the phosphorylated C-terminal region and the cytoplasmic face of the TM region is indicated by the perturbation of the methionine methyl resonances on the TM region, as follows (page 12, lines 3-11 and 14-17).

“The cross-saturation experiments revealed that either the TM region of β_2 AR or lipids adhere to the phosphorylated TM-proximal region (Fig. 4). Furthermore, we also observed the perturbation of the resonances from M215^{5.54} and M279^{6.41}, which are located in the TM region, upon phosphorylation (Fig. 3). It is unlikely that the interaction between the phosphorylated C-terminal region and lipids induces the observed perturbation of the resonances from M215^{5.54} and M279^{6.41}, because the methyl groups of M215^{5.54} and M279^{6.41} are buried in the TM helix bundle and $> 7 \text{ \AA}$ away from the lipid atoms. In addition, the cytoplasmic face of the TM region of β_2 AR is positively charged (Supplementary Fig. 5a), which would complement the negatively charged phosphate groups incorporated within the C-terminal region.”

“Thus, it is most likely that the phosphorylated TM-proximal region interacts with the cytoplasmic face of the TM region by an electrostatic interaction, and the interaction alters the microenvironments around M215^{5.54} and M279^{6.41}.”

Comment1-3

2. In the Abstract, the authors conclude that they “found that these structures induced by the phosphorylation are preferable for the interaction of β_2 AR with β -arrestin.” The direct implication is that the structures induced are preferable relative to the non-phosphorylated forms with β -arrestin. However, the measurements in Figure 5 that describe the arrestin interactions are only for β -arrestin in the phosphorylated state. Measurements of the non-phosphorylated receptor + β -arrestin are not reported. The authors cannot rule out that a large change in chemical shifts of the Met residues would be observed upon addition of β -arrestin to the non-phosphorylated receptor (containing full agonist) without running the experiment. It may be that the full agonist allows an induced fit of β -arrestin.

According to the reviewer’s comment, we examined the binding between unphosphorylated β_2 AR and β -arrestin 1, in the presence of the full agonist, by the surface plasmon resonance (SPR) method, as shown in the following Figure.

SPR analyses of the interaction between β -arrestin 1 and β_2 AR embedded in rHDLs.

Overlay of the sensorgrams obtained for the interaction between 31 ~ 500 nM β -arrestin 1 and immobilized unphosphorylated (left) or phosphorylated (right) β_2 AR embedded in rHDLs, in the full agonist-bound state.

As a result, no responses were observed upon the addition of β -arrestin 1 to unphosphorylated β_2 AR, whereas the responses were robustly observed in the case of the phosphorylated β_2 AR. These results suggested that the unphosphorylated β_2 AR does not bind to β -arrestin, which is in good agreement with the previous report (Gurevich *et al.*, *Pharmacol. Ther.* (2012) **133**, 40-69, Ref. 26).

In the revised manuscript, we added the above figure as Supplementary Fig. 4 and the following sentence about the SPR experiments, on page 10, lines 13-15.

“Surface plasmon resonance (SPR) analyses revealed that a β_2 AR mutant binds to β -arrestin 1 in a phosphorylation-dependent manner (Supplementary Fig. 4), in good agreement with the previously reported interaction between GPCRs and arrestins²⁶.”

In the Abstract section, we changed the sentence so that the conclusion can be directly derived from the comparison of the resonances from M215^{5.54}, as follows (on page 2, lines 10-11).

“In addition, we found that the conformation induced by the phosphorylation is similar to that corresponding to the β -arrestin-bound state.”

Comment 1-4

3. In the Abstract, the authors conclude that “Our analyses revealed that.... secondly, the interaction induces the spatial rearrangement of the transmembrane helices toward the conformation that corresponds to the β -arrestin-bound state.” This conclusion is based on the observation of two Met side chain methyl group resonances (Met5.54 and Met6.41). Actually, the conclusion is based on only one Met side chain methyl chemical shift (Met5.54) since the Met6.41 chemical shift in the phosphorylated receptor is not observed. (It might be that this resonance overlaps with that of Met2.53 or is shifted in a different direction.) The single Met5.54 chemical shift ($\Delta 13C \sim 0.1$ ppm) does not directly report on the spatial rearrangement of the transmembrane helices. For the non-NMR reader, this conclusion would be very misleading.

As in our previous NMR study of another GPCR (Okude et al., *Angew. Chem. Int. Ed.* 2015, Ref. 29), the chemical shifts of the methionine methyl groups in the TM region sensitively reflected the conformational change of the transmembrane helices. In the case of β_2 AR, the 1H chemical shift of M215^{5.54} is determined by the ring current effect of the neighboring F282^{6.44}, and thus it is considered to be a sensitive probe to monitor the spatial rearrangement of TM6 (Supplementary Fig. 6).

Supplementary Figure 6. Conformational change around M215^{5.54} and TM6 upon activation. The crystal structures of β_2 AR with an inverse agonist, carazolol (PDB ID: 2RH1)⁸ (grey) and with a full agonist, BI-167107, and a G-protein (PDB ID: 3SN6) (purple) are overlaid and shown in a cytoplasmic view. TM helices are depicted by $C\alpha$ traces, and

M215^{5.54} and F282^{6.44} are depicted by stick models. In the inverse agonist-bound state, the aromatic ring of the F282^{6.44} side chain is farther away from the methyl group of the M215^{5.54} side chain. In contrast, in the full agonist-bound state, the aromatic ring of the F282^{6.44} side chain approaches the methyl group of the M215^{5.54} side chain, and TM6 moves outward. The structural model was prepared with Cuemol (<http://www.cuemol.org/>).

In the revised manuscript, we described that the methionine methyl resonances are sensitive to the spatial rearrangement of the neighboring helices in the Discussion section, as follows (on page 12, line 18-page 13, line 6).

“As in our previous NMR study of another GPCR²⁹, the chemical shifts of the methionine methyl groups in the TM region sensitively reflected the conformational change of the TM helices. M215^{5.54} and M279^{6.41}, which were perturbed upon phosphorylation in the NMR spectra of β_2 AR, are located on the intracellular halves of TMs 5 and 6, respectively, and the ¹H chemical shifts of M215^{5.54} and M279^{6.41} are affected by the ring current effects from F282^{6.44} and Y219^{5.58}, respectively, which are sensitive to the spatial arrangement of the intracellular halves of TM5 and TM6, a possible arrestin binding site (Supplementary Fig. 6). These chemical shift perturbations of the methionine residues suggest that the intracellular halves of TM5 and TM6 become spatially rearranged upon phosphorylation.”

We also revised the manuscript so that the conclusions in the Abstract and Results sections are directly derived from the experimental data. Firstly, we removed the description in the Results section about the spatial rearrangement of the transmembrane helices (on page 9, line 1). Secondly, we changed the sentence in the Abstract section on page 2, lines 8-10, as follows.

“the phosphorylation of the C-terminal region allosterically alters the conformation around M215^{5.54} and M279^{6.41}, located on transmembrane helices 5 and 6, respectively.”

Comment 1-5

The issue I have with the manner in which the authors state their conclusions continues in the Discussion section. In a general way, the use of the term “Therefore” in my opinion is the final statement in a logical argument (either through deductive or inductive reasoning). In a deductive argument, if one accepts the premises then the conclusion must be true (i.e. a simple form would be “If A, then B. One observes A. Therefore, B.) In an inductive argument, the conclusion is not

necessarily true. An example is that one concludes after observing ravens for several days that “All ravens are black”. It may well be the case that there is a raven that is white that was missed.

I agree with the reviewer's comment. According to the reviewer's suggestion, we carefully changed the descriptions throughout the Discussion section.

Comment 1-6

Paragraph 2 of the Discussion is an example (the first of six instances of using the term “Therefore” in the Discussion).

The cytoplasmic face of the TM region of β 2AR is positively charged (Supplementary Fig. 5a), whereas the lipids in rHDLs and the phosphate groups attached to the C-terminal region are negatively charged. Therefore, the observed cross-saturation effect is likely to correspond to the electrostatic interaction with the cytoplasmic face of the TM region.

The cross saturation effect may well correspond to a positive-negative interaction. However, an explanation that would cover BOTH the broadening of the amide resonances upon phosphorylation and the cross saturation effect would be that the negative-negative interaction (C-terminus and phosphate groups) leads to a ensemble of conformations (i.e. there is no single defined state as their might be with complementary interactions) and hence broadening. The cross-saturation of the TM proximal residues (vs. the TM-distal) residues is just because they are closer.

One might say that the positive-negative complementarity between the phosphate groups and the cytoplasmic surface of the receptor suggests that these regions associate and would be consistent with the cross saturation results.

I appreciate the reviewer's comments about the possibility that the line broadening and cross-saturation reflect the negative-negative interaction between the phosphate groups of the phospholipids and the phosphorylated C-terminal region. According to the reviewer's comments, we added a description concerning the observed line broadening, as follows (on page 11, line 19-page12, line 2):

“Upon phosphorylation, the amide and methyl resonances derived from the TM-proximal region were broadened (Fig. 2b-d). The line broadening suggests that the phosphorylated TM-proximal region interconverts between multiple states, which are probably due to the negative-negative charge interaction between the phosphorylated TM-proximal region and lipids.”

In addition, we added a description about the cause of the cross-saturation, as described in the responses to comment 1-2 (on page 12, lines 3-11 and 14-17).

Comment 1-7

In short, I think the approach taken and the data presented are very strong and merit publication in Nature Communications. The authors simply need to clearly distinguish (particularly in the Title and Abstract) what can directly be concluded from the data and what one might speculate based on these conclusions.

I thank the reviewer for carefully reading our manuscript. In the revised manuscript, we now carefully distinguish what can be directly concluded from the experimental data and what one might speculate based on the conclusions. We also changed the manuscript title on page 1, lines 2-3, as follows:

“Phosphorylation-induced conformation of β_2 -adrenoceptor related to arrestin recruitment revealed by NMR”

Reviewer #2

Comment 2-1

This manuscript describes an interesting study on the effects of phosphorylation on conformation of the beta2 adrenergic receptor. The authors utilize several NMR spectroscopy-based approaches to study the role that phosphorylation plays in managing receptor conformation and arrestin binding. In addition to methionine labeling, the authors exploit intein-based fusion technology to replace the receptor's C-terminus with an isotope-labeled carboxy terminus produced in bacteria. Upon reconstitution in a lipid bilayer and in vitro phosphorylation, the authors were able to discern specific conformational changes in the TM region as well as in the C-terminus following phosphate incorporation and arrestin binding. These are novel approaches applied to G protein-coupled receptors and novel findings.

Although this is a very interesting study there are several issues that the authors need to address in their manuscript, preferably either through additional experimentation or minimally through detailed, extensive discussion.

Comments:

The cross saturation results are quite interesting. Some reference to the highly conserved basic regions of GPCRs around the cytosolic faces of TM5 and TM6 should be made. Hundreds of reports focusing on these basic regions have been published throughout the years but a strong consensus of their role in receptor function has not been met. Structural evidence, particularly the GPCR-G protein complex structures have provided little role for these regions in G protein coupling. The interaction of the phosphorylated C-terminal tail, as suggested in this current study, could finally provide some rationale for their conservation.

I agree with the reviewer's comment. As the reviewer pointed out, the conservation of the basic regions of GPCRs is often focused in reports and reviews, such as Ballesteros *et al.* (*Mol. Pharmacol.* (2001) **60**, 1-19) and Lu *et al.* (*Trends Pharmacol. Sci.* (2002) **23**, 140-146). In the revised manuscript, we cited these reviews as Refs. 38 and 39, and added a sentence about the conserved basic region, as follows (on page 15, lines 11-14).

"The basic regions around the cytoplasmic faces of TM5 and TM6 are reportedly highly conserved among many GPCRs^{38,39}, and the crystal structures of GPCRs other than β_2 AR also contain positively charged clusters on the cytoplasmic face of the TM region

(Supplementary Fig. 5b-d), although their role in receptor function is not well understood.”

Comment 2-2

Also, it should be noted that the original crystal structure of rhodopsin (Palczewski et al 2000) reported a density in the intracellular core (where the G protein and arrestin binds) that was modelled as the receptor's C-terminus. Granted this was a crystal structure however the receptor was obtained from native sources (not recombinant) and was not treated with a phosphatase. Limited, or partial phosphorylation the rhodopsin C-terminus can not be ruled-out in this preparation.

I agree with the reviewer's suggestion. In the revised manuscript, we cited Palczewski et al. (*Science* (2000) **289**, 739-745) as Ref. 28, and added the following sentence (on page 12, lines 11-14).

“In the original crystal structure of rhodopsin from a native source²⁸, an electron density was observed in the intracellular core, and it is plausible that the density reflects the interaction between the phosphorylated C-terminal region and the TM region.”

Comment 2-3

The reference to- and discussion of the beta2AR-arrestin study by electron microscopy (and functional analyses) should be tempered in that the receptor used in the study was a C-terminal fusion with the V2R receptor. The V2R is a class B set of receptors that displays sustained interactions with arrestin. It's interaction with arrestin will be different than with class A receptors like the beta2AR. This is an artificial system that was used to stabilize a complex between arrestin and the beta2AR. This is not the system used in this current study and therefore comparisons should not be made. Note that the class designation should not be confused with the Family A/B/C/D/E/F GPCR nomenclature and only refers to the interaction with arrestin. The reference to these beta2AR-arrestin structure was made during the discussion of the potential contribution of the phospholipid bilayer under which these current experiments were performed. Indeed, it is likely that the lipid environment plays a critical role, not tested previously in this manner, but the comparison with the previously reported EM work should not be made.

I agree with the reviewer's comment. According to the reviewer's suggestion, we

removed the description concerning the comparison with the previously reported EM work in the revised manuscript and Supplementary Information.

Comment 2-4

The phosphorylation-dependent movement of M215 is quite interesting. This, coupled with the chemical shift observed on M279, suggests that phosphorylation influences the conformation of the TM regions of the receptor. Although the authors provide a detailed discussion of the phenomena the authors do not address the mechanistic relevance. Inward movement of TM6, as observed following phosphorylation, would normally be associated with the inactive conformation of the receptor. In contrast, as the authors know, is associated with the outward movement of TM6, a form that can be stabilized by G proteins, conformation-selective antibodies, and arrestins. Some discussion of the discrepancy must be made.

The inward movement of TM6 upon phosphorylation, proposed in this study, does not mean that the full agonist-bound β_2 AR adopts an inactive conformation upon phosphorylation. In the active state, it is conceivable that the degrees of the outward shift of TM6 varied from 3.5 Å to 14 Å, based on a comparison of the crystal structures of GPCRs in various active states, as shown in the following Figure.

The degrees of the outward movement of TM6 in various active states. The outward movements of TM6 were measured at Glu^{6.30} C α between the inactive- and active-conformation structures. The inactive state structures are depicted by grey tubes⁸⁻¹⁰. The active state structures of adenosine A_{2A}¹¹, rhodopsin bound to arrestin 1¹², β_2 AR bound

to Nb6B9¹³, and β_2 AR bound to G-protein² are depicted by cyan, green, orange, and red tubes, respectively. The C α carbons of Glu^{6,30} are depicted by spheres. The PDB IDs of the structures are as follows: inactive adenosine A_{2A}: 3EML, active adenosine A_{2A}: 3QAK, inactive rhodopsin: 1F88, active rhodopsin bound to arrestin 1: 4ZWJ, inactive β_2 AR: 2RH1, active β_2 AR bound to Nb6B9: 4LDE, active β_2 AR bound to G-protein: 3SN6.

We thus supposed that the proposed inward movement of TM6 upon phosphorylation is included in the categories of the active conformations. Particularly, in the crystal structure of the arrestin 1-rhodopsin complex, TM6 exhibited a smaller outward shift than that of the Gs- β_2 AR complex. The difference suggested that the proposed inward shift of TM6 upon phosphorylation may be suitable for the phosphorylation-dependent arrestin recruitment.

In the revised manuscript, we added the above figure as Supplementary Fig. 7. We also added the discussion about the mechanistic relevance of the proposed inward shift, as follows (on page 13, line 20-page 14, line 7).

“The crystal structures of GPCRs revealed that the degrees of the outward movement of TM6 vary among GPCRs in the active states, from 3.5 Å to 14 Å (Supplementary Fig. 7). The proposed inward movement of TM6 upon phosphorylation should be included in the categories of these active conformations. Particularly, in the structure of rhodopsin bound to arrestin, the degree of the outward movement of TM6 is smaller than that in the structure of β_2 AR bound to G-protein, suggesting that arrestin recognizes a GPCR that exhibits a relatively small outward movement of TM6. The proposed inward movement of TM6 upon phosphorylation would thus be suitable for the phosphorylation-dependent arrestin recruitment.”

Comment 2-5

It would be even better if the authors provided biochemical or biophysical data that tested the capacity of the G proteins or conformation-selective antibodies to couple to the phosphorylated receptor. For example, does agonist binding change in a phosphorylation-dependent manner is this preparation? Inward movement of TM6, or stabilization of the inactive conformation, should decrease agonist binding affinity. Do G proteins or conformation-selective antibodies (nanobodies) influence the chemical shifts observed on the phosphorylated receptor? The nanobodies (eg Nb80) in this case would be quite powerful as they should be able to bind the phosphorylated receptor.

I thank the reviewer for the fruitful suggestion that the capacity of G-proteins or conformation-selective antibodies to couple to the phosphorylated receptor would provide clear insights into the mechanistic relevance of the proposed conformational change upon phosphorylation. According to the reviewer's suggestion, we examined the capacity of the nanobody Nb6B9, which selectively recognizes the intracellular face of β_2 AR in the active state with a large outward movement of TM6, by SPR experiments, as shown in the following figure.

SPR analyses of the interaction between Nb6B9 and β_2 AR embedded in rHDLs.

Sensorgrams obtained for the interaction between 4 ~ 1,000 nM Nb6B9 and immobilized unphosphorylated (black) or phosphorylated (Red) β_2 AR embedded in rHDLs, in the full agonist-bound state.

The SPR analyses revealed that the binding of Nb6B9 to β_2 AR was decreased upon phosphorylation. This result is consistent with the inward movement of TM6 upon phosphorylation that we proposed from the NMR data.

In the revised manuscript, we added the SPR data as Supplementary Fig. 8 and a description of the experiment, as follows (on page 14 lines 8-13).

“To validate the proposed inward movement of TM6 upon phosphorylation, the capacity of the nanobody Nb6B9 was examined by SPR methods. Nb6B9 selectively recognizes the intracellular face of β_2 AR in the active state, with a large outward movement of TM6³⁵. As a result, the responses observed for Nb6B9 binding to the phosphorylated β_2 AR were lower

than those to the unphosphorylated β_2 AR (Supplementary Fig. 8), which is consistent with the proposed phosphorylation-induced inward movement of TM6.”

Comment 2-6

In the discussion section (lines 258-268) the authors state that the “results clearly suggested that phosphorylation of the C-terminal regions induced the conformational changes dependent on the position of the phosphorylated residues”, in reference to the phosphorylation bar code hypothesis. The term ‘clearly’ is quite strong here based on the data provided here.

I agree with the reviewer’s comment. According to the reviewer’s suggestion, we removed the term ‘clearly’ in the revised manuscript.

Comment 2-7

Finally, several studies have been reported on the structural analysis of agonist-bound GPCRs, using crystallography, EPR and NMR spectroscopy. In each case it was difficult to observe, from a population perspective, stabilization of the active conformation with an agonist alone (ie. without a G protein or conformation-selective antibody). The fact that GRK-mediated phosphorylation occurs in an active receptor-dependent manner, suggests that the result of GRK interaction, phosphorylation, should also maintain the receptor in an active conformation. This would make some sense since it would phosphorylation enhances arrestin recruitment both the C-terminus and to the receptor core. Some discussion of these properties in the context of the experimental results of this study should be made.

I thank the reviewer for this suggestion. According to the reviewer’s suggestion, we added a description, as follows (on page 13, lines 10-14).

“Recent structural studies indicated that G proteins or conformation-selective nanobodies modulate the conformation of the TM region from the intracellular side³⁰⁻³⁴. The proposed phosphorylation-induced conformational change of the TM region could be considered as a similar phenomenon. This is reasonable, since arrestin recognizes both the phosphorylated C-terminal region and the TM region.”

Comment 2-8

Minor: the cartoon of the phosphorylated residues interacting with the basic region (of TM5 and

TM6) should be modified. It is a little misleading to suggest that the residues are as deep into the cytosolic core of the receptor where the G proteins and arrestin bind.

I agree with the reviewer's suggestion. In the revised manuscript, we modified the positions of the basic region and the phosphorylated residues in Fig. 6, as follows.

Figure 6. Role of the phosphorylation of the C-terminal region in signal transduction.

In the unphosphorylated state, the C-terminal region of $\beta_2\text{AR}$ is unstructured. Upon phosphorylation by GRKs, the TM-proximal region of the C-terminal region of $\beta_2\text{AR}$ adheres to the cytoplasmic face of the TM region, and the intracellular halves of the TM helices become spatially rearranged, which preferentially activates arrestin-mediated signal transduction. The adhesion of the phosphorylated C-terminal region to the TM-region and the spatial rearrangement of the TM helices would generate the structural motif for β -arrestin binding.

REVIEWERS' COMMENTS:

Reviewer #1 (Remarks to the Author):

The authors have fully addressed my concerns. As I wrote in my original review, this work advances the field and merits publication.